# SHARED MODULAR RECURRENCE FOR UNIVERSAL MORPHOLOGY CONTROL

## ABSTRACT

A universal controller for any robot morphology would greatly improve computational and data efficiency. By utilizing *contextual* information about the properties of individual robots and exploiting their modular structure in the architecture of deep reinforcement learning agents, steps have been made towards multi-robot control. When the robots have highly dissimilar morphologies, this becomes a challenging problem, especially when the agent must generalize to new, unseen robots. In this paper, we hypothesize that the relevant contextual information can be partially observable, but that it can be inferred through interactions for better multi-robot control and generalization to contexts that are not seen during training. To this extent, we implement a modular recurrent transformer-based architecture and evaluate its (generalization) performance on a large set of MuJoCo robots. The results show a substantial improved performance on robots with unseen dynamics, kinematics, and topologies, in four different environments.

## 1 INTRODUCTION

Reinforcement Learning (RL) has shown to be very promising for robotic control (Levine et al., 2016; Kalashnikov et al., 2018; Andrychowicz et al., 2020). In an effort to close the gap with real-world applications, a lot of work has focussed on RL agents that are able to generalize control to different tasks, e.g. manipulating different objects or acting in different environments. Recently, large datasets of robot trajectories have been established and are being used to train such generalizable agents by learning from demonstrations in an offline fashion with foundation models (Brohan et al., 2022; Zitkovich et al., 2023; Vuong et al., 2023). Several works show promising possibilities for a single model to adapt not only to different scenes and goals, but also to different embodiments that the model has seen demonstrations of (Doshi et al., 2024; Octo Model Team et al., 2024). Zero-shot generalization to robots that were not seen during training, however, remains very challenging.

Different robots may be suitable for various different tasks and environments. The UNIMAL design space (Gupta et al., 2021) was developed for the evolution of diverse robots that can perform varied tasks. Different robot morphologies can namely have advantages over each other, dependent on the goal. The UNIMAL design space contains more than 1000 different robots, a small subset of which is shown in Figure 1. It is infeasible to train a policy from scratch for every robot: training (or even fine-tuning) a policy for every new robot that we are interested in requires expensive compute, excessive use of data, and often tedious hyperparameter tuning. A universal controller that can generalize control to any robot morphology could drastically improve efficiency. The dataset of robots in the UNIMAL design space is suitable for the development and evaluation of RL agents that can generalize control to any robot.

By framing this problem as a multi-task RL (Vithayathil Varghese & Mahmoud, 2020) problem, each robot can be considered a new task that has some specific *contextual* features, such as the mass of different limbs and the topology of the robot. The goal in this multi-task setting is to learn a universal controller that can control any robot on basis of its observations and context. This is not only challenging because robots can have different action and state spaces, but also because robots with different morphologies and/or dynamics might learn tasks in different ways. The multi-task framework allows us to evaluate the performance of an agent during multi-robot training, and its zero-shot generalization performance (Kirk et al., 2023) to new, unseen robots.

Previous work on universal morphology control assumes that the contextual features that describe properties of the robot are fully observable. In this paper, we hypothesize that those features are arbitrarily available and do not necessarily encapsulate enough information to be able to represent the true context needed for optimal (generalizable) control. We build upon previously found effective modular architectures for robotic control (Gupta et al., 2022; Xiong et al., 2023), and address the essentially unobservable context with a recurrent block, while retaining the system's modularity. This paper empirically shows that the proposed system improves multi-robot control on varied training morphologies, and enables better zero-shot generalization to unseen test morphologies.

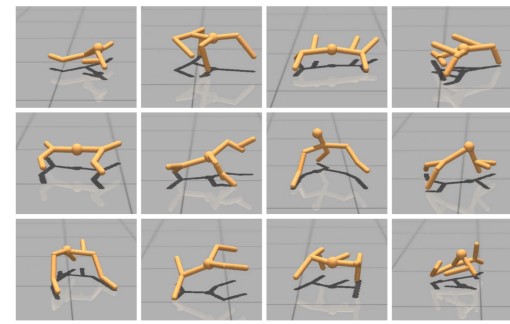

Figure 1: Example of robots that can be found in the UNIMAL design space (Gupta et al., 2021).

## 2 BACKGROUND

### 2.1 CONTEXTUAL MARKOV DECISION PROCESS

Here, the problem of learning RL policies that are trained on a set of training robots and must generalize to unseen test robots is considered. This problem can be formulated as a Markov Decision Process (MDP) where the agent can only partially access the MDP during training. An MDP is a tuple $(\mathcal{S}, \mathcal{A}, \mathcal{T}, \mathcal{R}, \rho)$, with state space $\mathcal{S}$, action space $\mathcal{A}$, transition function $\mathcal{T}(s_{t+1}|s_t, a_t)$, mapping current state $s_t \in \mathcal{S}$ and action $a_t \in \mathcal{A}$ to a probability distribution over next states $s_{t+1} \in \mathcal{S}$, reward function $\mathcal{R} : \mathcal{S} \times \mathcal{A} \rightarrow \mathbb{R}$ and initial state distribution $\rho(s_0)$. A Contextual Markov Decision Process (CMDP (Hallak et al., 2015)) is an MDP where states can be decomposed $s_t = (s'_t, c)$, into the underlying state $s'_t \in \mathcal{S}'$ and a context $c \in \mathcal{C}$. The context is sampled at the start of each episode and remains static until the episode ends. In the current environment, the context defines the robot to be controlled. This framework allows to evaluate zero-shot generalization, that is, generalization to contexts not seen during training, by defining a training and testing set of robots (Kirk et al., 2023).

The UNIMAL design space contains modular robots (Gupta et al., 2021) that are simulated with the MuJoCo physics engine (Todorov et al., 2012). Such robots consist of a set of nodes, or limbs, that share the same state space and action space, i.e. $\mathcal{S} = \{\mathcal{S}^i | i = 1, ..., N\}$ and $\mathcal{A} = \{\mathcal{A}^i | i = 1, ..., N\}$ for robots with $N$ limbs. Besides the underlying states, a node-level context $\mathcal{C} = \{\mathcal{C}^i | i = 1, ..., N\}$ is provided in the state that describes information of the limbs (e.g. mass, initial position with respect to the parent limb, and the initial position of each joint attached to the limb). The node-level observation and context features that are provided to the agent are listed in Table 1 in Appendix A. Previous methods exploited this modular structure in combination with modern architectures for effective multi-robot control (Huang et al., 2020; Kurin et al., 2021; Gupta et al., 2022; Xiong et al., 2023). Rather than aiming to develop context-agnostic agents, Gupta et al. (2022) and Xiong et al. (2023) condition the agent on the available contextual information with methods coined *MetaMorph* and *ModuMorph*, respectively. In doing so, they did not only show improved performance during training, but also on generalization to unseen robots. However, the gap between the performance on training and testing robots remains substantial.

### 2.2 PARTIALLY OBSERVABLE CONTEXT

In this paper, we recognize and exploit the fact that relevant contextual information can be partially observable. The features that are provided, namely, depend on what information is available and whether they can be effectively processed in the agent architecture. A lot of (possibly) relevant contextual information is not available, such as the friction and damping of limbs. Additionally, the true adjacency matrix that describes the organization of the limbs can not be provided effectively to an agent with a modular architecture due to complications with positional encoding, as pointed out by Xiong et al. (2023). Therefore, topological information is provided rather implicitly instead,

through the position of a limb with respect to its parent limb. Lastly, we do not have an adequate way of providing the agent with more abstract features, such as the influence of different limbs on each other during movement. It is only through interaction with the environment that the agent can infer such features to perform optimally in the current task.

Although the underlying state is considered to be fully observable for one MuJoCo robot (Todorov et al., 2012), the CMDP could be partially observable (Ghosh et al., 2021). In this setting, the emission or observation function defined in partially observable MDPs (Spaan, 2012), $\phi : \mathcal{S} \to \mathcal{O}$, maps a state to an observation $o_t \in \mathcal{O}$ that contains the underlying state and the observable context, $c^+$, for every time step $t$: $o_t = \phi((s'_t, c)) = (s'_t, c^+)$. This would not be a problem when we only want to learn control for a set of robots seen during training; with enough representational power, the agent can simply overfit. The challenge arises when the agent encounters robots with new contextual features. In this case, it can be more effective to use experience collected during an episode, as contextual features (required for better/generalizable control) can be encapsulated in or related to this dynamical information. Since the underlying state of MuJoCo robots is widely considered fully observable, any inferred unobservable state information *must* be from the missing context. Incorporating memory into the agent architecture (Hausknecht & Stone, 2015) can allow the agent to (implicitly) infer and quickly adapt to necessary unobservable contextual information. Such memory mechanism must preserve the modular structure of the agent's architecture to remain compatible with the multi-robot setting in which robots can have a different number of limbs.

## 3 RELATED WORK

### 3.1 UNIVERSAL MORPHOLOGY CONTROL

We build upon previous work aimed at learning an RL policy that can generalize control to different robot morphologies, even when state and action dimensionalities can change. Effective approaches utilized the modularity in robots (Pathak et al., 2019) and introduced weight sharing across different modules (Huang et al., 2020). Several works adopted graph neural networks (Scarselli et al., 2008; Wang et al., 2018) or transformers (Vaswani et al., 2017; Kurin et al., 2021) as inductive biases to more explicitly infer relationships between different limbs or modules through message-passing. More recently, multi-robot training has been evaluated on a larger scale of different morphologies with the introduction of the UNIMAL design space (Gupta et al., 2021). Gupta et al. (2022) constructed training and testing sets of robots to evaluate multi-robot control, and showed the effectiveness of a modular transformer-based approach when contextual information is provided to the agent. By further exploiting this contextual information in the agent's architecture, Xiong et al. (2023) demonstrated improved training and generalization performance.

Steps towards robotic applications in the real world also exploit such modular transformer-based architectures for generalizable and transferrable control. Several recent works incorporate additional information about the robot topology through a graph encoding or sparse attention matrices (Patel & Song, 2024; Sferrazza et al., 2024) for improved generalization in simulation and the real world. Fine-tuning and policy distillation approaches have also been proposed for generalization purposes (Przystupa et al., 2025; Xiong et al., 2024). Lastly, more complex modular architectures were introduced for effective transfer to unseen robots in simulation and the real world (Bohlinger et al., 2024; Li et al., 2024), although evaluating on smaller sets of, and relatively similar robot morphologies. None of these approaches consider this generalization problem as partially observable. Earlier works did suggest that "implicit system identification" with memory-based policies can benefit robotic control, but did not utilize a modular system, nor evaluate generalization on varied morphologies (Yu et al., 2017; Peng et al., 2018). We are the first to show that added modular recurrence can improve zero-shot generalization on a diverse set of robot morphologies.

### 3.2 NEURAL ARCHITECTURES

The effectiveness of the transformer architecture (Vaswani et al., 2017) for multi-robot control, lies in its capability to model pairwise dependencies between limbs with self-attention. Self-attention can be defined as $A = \sigma(QK^T/\sqrt{d})V$, with query, key and value matrices $Q, K, V \in \mathbb{R}^{N \times d}$, for robots with $N$ limbs and a hidden size of $d$. Learnable parameters $W_Q, W_K$ and $W_V$ map the input $X \in \mathbb{R}^{N \times d}$ to those matrices, i.e. $Q = XW_Q$, $K = XW_K$ and $V = XW_V$, and $\sigma(\cdot)$ is a row-wise

softmax function. In addition to the attention mechanism, Xiong et al. (2023) utilize hypernetworks (Ha et al., 2017) to more explicitly condition the agent on the available node-wise contextual information. Briefly, they train a hypernetwork that is conditioned on the observable context to generate (1) the parameters of a node-wise encoder that produces $X_V$ with which the value in the subsequent transformer encoder layer is calculated as $V = X_V W_V$, (2) $X_Q$ and $X_K$, where the query and key matrices are now defined as $Q = X_Q W_Q$, $K = X_K W_K$, respectively, and (3) the parameters of a node-wise decoder that projects the output of the transformer encoder.

Recurrent neural networks (RNNs) like LSTMs (Hochreiter & Schmidhuber, 1997) are useful architectures to deal with partially observable domains in deep RL. Tang & Ha (2021) combined LSTMs with attention to develop systems that can adapt to changes (permutations) of the input. Here, we utilize a variation of this modular architecture to investigate its effectiveness for multi-robot control and zero-shot generalization.

### 3.3 Zero-shot Generalization and Meta Reinforcement Learning

In this paper, we look into zero-shot generalization of the learned policies (Kirk et al., 2023): the aim is to learn a policy that performs well on a set of training robots, and in particular can control unseen test robots without additional learning in the test environments. Nonetheless, one could also interpret access to the preceding trajectory as few-shot adaptation to the test environment, similar to the meta-learning approach RL$^2$ (Duan et al., 2016). In this light, gradient-based meta-learning methods like MAMAL (Finn et al., 2017) could be employed to adapt the network after every transition to adapt the policy to the unseen test environment. However, this would slow down execution considerably in comparison to RNN inference. Another option would be in-context learning, where agents have to generalize to new tasks without updating parameters, but can utilize preceding trajectories (see Moeini et al., 2025, for a recent survey), particularly using transformers (Laskin et al., 2023). This would be in practice very similar to using an RNN, as the transformer would also have access to the agent's history. However, here too, RNNs are faster during execution as they are linear in the length of the history, instead of quadratic like transformers (Vaswani et al., 2017).

## 4 Shared Modular Recurrence

### 4.1 Recurrent PPO

In the current multi-task RL problem, we want to find a universal control policy that is effective for any robot we can encounter in the UNIMAL design space by only training on a set of $K$ training robots. One effective approach to RL in partially observable domains, is to learn an encoding of the belief over the agent's true state. This is often done by, at every time step $t$, encoding the action-observation history (AOH) $\tau_t^k = (o_0^k, a_0^k, \ldots, o_{t-1}^k, a_{t-1}^k, o_t^k)$ with an RNN, for (in the current domain) robot $k$ the agent is controlling. In this way, the training objective can be formulated as finding parameters $\theta$ for policy $\pi_\theta(a_t^k|\tau_t^k)$ that maximize the (discounted) cumulative reward, averaged over all training robots: $\max_\theta \frac{1}{K} \sum_{k=1}^{K} \mathbb{E}_{\pi_\theta}[\sum_{t=0}^{H} \gamma^t r_t^k]$, with task horizon $H$. We implemented a recurrent version of Proximal Policy Optimization (PPO) (Schulman et al., 2017) to optimize this objective.

Recurrent experience replay (Kapturowski et al., 2018), originally developed for DRQN (Hausknecht & Stone, 2015), is implemented here for PPO to effectively sample from the roll-out buffer. In the current setting, episodes can namely be of varying lengths with a maximum of $H = 1000$ time steps. Parallel training on multiple complete trajectories therefore requires padding and can quickly saturate memory. This problem can be solved by storing overlapping chunks of episodes and using a burn-in period for the RNN during training, as introduced by Kapturowski et al. (2018) for Deep Recurrent Q-Networks (Hausknecht & Stone, 2015). The hidden states at the beginning of each chunk are stored and used at the start of the burn-in period. Here, a chunk size of $m = 80$ and a burn-in period of $l = 20$ will be used, as those values were reported by Kapturowski et al. (2018) to be effective.

## 4.2 SHARED RECURRENT NETWORK

Normally, a single recurrent block is introduced in the agent's architecture to encode global actions and observations. To retain modularity, however, we cannot encode the global AOH. Instead, we adopt and adapt a recurrent architecture that processes components of the input separately (Tang & Ha, 2021): every limb-level action and observation are processed individually through an RNN to encode local AOHs $\tau_t^i = (o_0^i, a_0^i, \ldots, o_{t-1}^i, a_{t-1}^i, o_t^i)$ for every limb $i$ (we omit the superscript that indicates the robot as our policy is controlling only one robot at a time). Since nodes share the same state space, the parameters of this RNN are shared to increase the scalability of this approach. We only keep track of individual hidden states $h_t^i = \text{RNN}(o_t^i, a_{t-1}^i, h_{t-1}^i)$ that encode the local AOH $\tau^i$, which are initialized with zeros. In this way, the agent can approximate the relevant history for every limb individually. There are various ways in which this modular recurrence can be incorporated in the agent's architecture. Here, we build upon state-of-the art architectures for universal morphology control: MetaMorph (Gupta et al., 2022) and ModuMorph (Xiong et al., 2023).

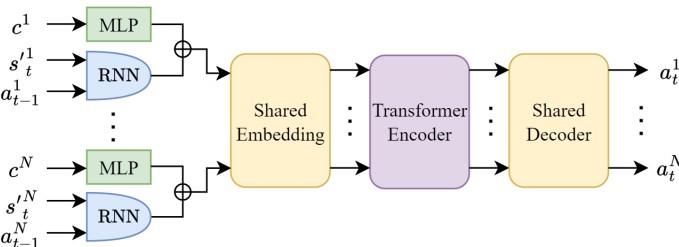

Figure 2: Illustration of the recurrent MetaMorph (R-MeMo) architecture. Note that the underlying state $s_t'$ and context $c$ that form the observation $o_t$ are separated, as only the underlying state can change during an episode. The available context is therefore not encoded through the (shared) RNN.

**Recurrent MetaMorph.** MetaMorph exploited a modular transformer-based architecture in which limb-wise observations are processed through a shared embedding layer, before applying multi-head attention. The resulting embeddings are mapped to actions with a shared decoder. As shown in Figure 2, our recurrent version of MetaMorph (**R-MeMo**) first encodes the underlying state and previous action with an RNN (shared among all limbs) and separately processes the observable context, as this part of the observation remains constant throughout an episode. After adding these embeddings to form $\tau_t$, the rest of the architecture is identical to MetaMorph.

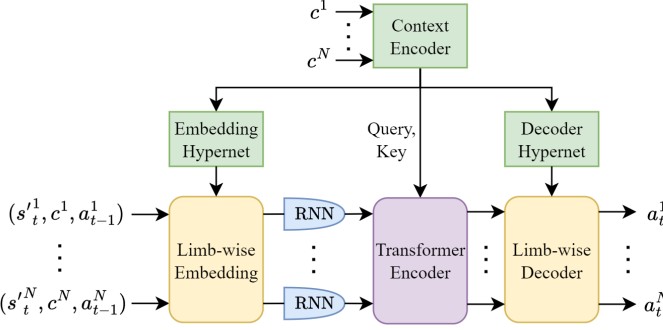

Figure 3: The ModuMorph architecture with added shared modular recurrence (R-MoMo).

**Recurrent ModuMorph.** ModuMorph was built on MetaMorph by introducing a hypernetwork that takes context as input and generates the parameters for the (now limb-wise) embedding layer, the query and key of multi-head attention, and the parameters for the (now limb-wise) decoder layer. Our recurrent ModuMorph version (**R-MoMo**), shown in Figure 3, uses the same hypernetwork and fixed attention as in the original architecture, but incorporates the shared RNN before the transformer

to approximate AOHs from latent encodings of the observation and previous action. For R-MoMo, the RNN could also be inserted before the embedding layer. This will, however, result in larger inputs to the embedding layer (if we keep the size of the hidden states the same): 128 instead of 54 per limb. For each limb, the embedding hypernetwork would then generate $(128 \times 128 =)$ 16384 parameters, instead of $(54 \times 128 =)$ 6912. In turn, the embedding hypernetwork itself would now have $(128 \times 16384 =)$ around 2M trainable parameters, instead of $(128 \times 6912 =)$ almost 900k. Thus, for a fair comparison with ModuMorph, we inserted the RNN after the embedding layer. In both architectures, a transformer receives the local (AOH) encodings from the RNN to infer relationships between different limbs. Introducing extra MLPs in MetaMorph and ModuMorph with a similar amount of parameters as the RNN (for a potential fair comparison) caused stability problems, as shown in Figure 10 in Appendix B, and was therefore avoided.

## 5  EXPERIMENTS

In this Section, experiments are performed with MetaMorph, ModuMorph, and their recurrent counterparts R-MeMo and R-MoMo, respectively. We use the MetaMorph version from Xiong et al. (2023), which they report to perform better than the original implementation.

### 5.1  EXPERIMENTAL SETUP

The training set of 100 robots, as constructed by Gupta et al. (2022), is used to train agents for multi-robot control. We first evaluate the agent's generalization performance on unseen variations of these training robots, where parameters that influence the dynamics and kinematics are altered (such as the damping of limbs or the angles joints can make). Subsequently, the performance on robots with unseen topologies is evaluated. The provided test set of robots with unseen topologies is randomly split into a validation (32 robots) and test (70 robots) set to experiment with different hyperparameters and evaluate generalization performance. We only validated two values for a regularization hyperparameter to select the models on which we report results. Xiong et al. (2023) namely found that this parameter can have a big impact on performance. All other hyperparameter values were taken from Xiong et al. (2023). See Appendix B for further details.

Agents are trained and evaluated in four different environments. In each of those, the agent has to maximize the robot's locomotion distance. In **Flat Terrain**, the agent needs to traverse a flat surface, while in **Incline** the robots are to be controlled on a surface that is inclined by 10 degrees. **Variable Terrain** contains a sequence of hills, steps and rubble, interleaved with flat terrain. Those sequences

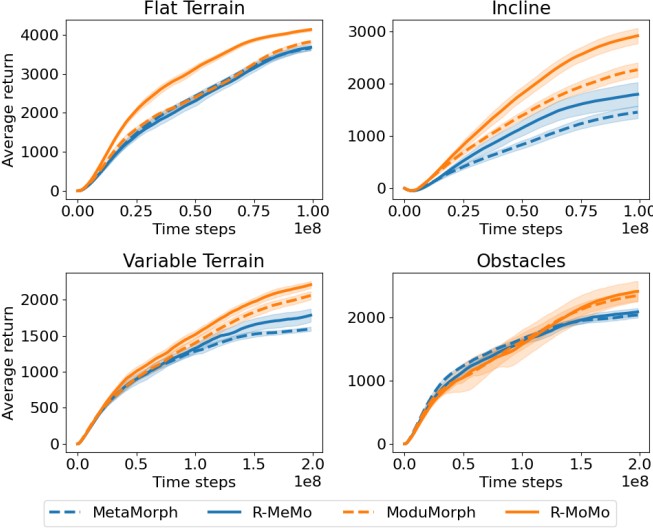

Figure 4: Training performance on the 100 training robots in the different environments. The average return with a 95% confidence interval over 10 seeds is visualized.

are randomly generated at the start of each episode. Finally, **Obstacles** is a flat terrain with randomly generated obstacles. In the latter two environments, the agent receives a 2D heightmap of its close surrounding, in addition to proprioceptive and contextual observations, to be able to react to changes in terrain. For more details on the environments, we refer to Gupta et al. (2021).

## 5.2 MULTI-ROBOT TRAINING PERFORMANCE

The training performance of the different methods on the 100 training robots, averaged over 10 seeds, is shown in Figure 4. Across all environments, the recurrent architectures (R-MeMo and R-MoMo) obtain either similarly high or higher returns than their non-recurrent baselines MetaMorph and ModuMorph. Particularly in the Incline environment, which is more difficult as dynamics play a more important role, modular recurrence results in better training performance. In general, ModuMorph seems to perform better than R-MeMo on the training robots, illustrating the effectiveness of the hypernetworks conditioned on the available context during multi-robot training. Due to the sequential processing of RNNs, the training time of R-MeMo and R-MoMo increases with respect to their baselines, which is reported in Appendix B. This is not the case during inference, as the agent then always has to process observations sequentially.

## 5.3 ZERO-SHOT GENERALIZATION TO DIFFERENT DYNAMICS AND KINEMATICS

Gupta et al. (2022) constructed a set of robots that have the same topologies as the training robots, but differ in a contextual feature, to evaluate zero-shot generalization to different dynamics or kinematics. For each training robot, they created four test robots with variations in armature, damping, gear, density, limb shapes or joint angles, resulting in a new set of 2400 test robots.

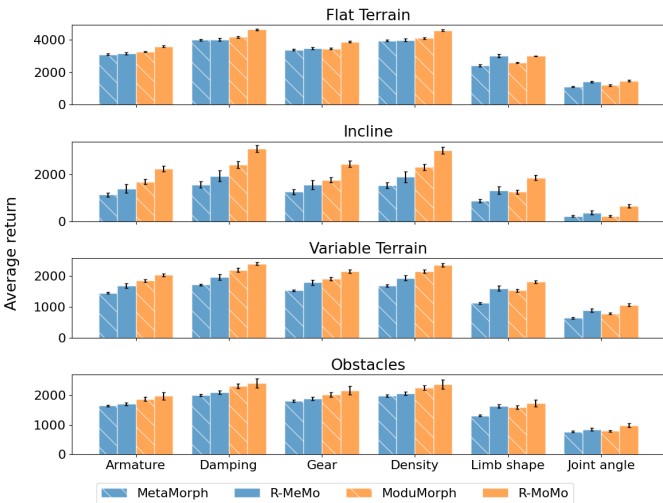

Figure 5: Test performance on training robots with changes in contextual features that result in different dynamics or kinematics. The average return with a 95% confidence interval over 10 seeds is reported.

Figure 5 shows the performance of the four evaluated methods on the test robots, for each of the changed dynamics and kinematics parameters. Over all changes, R-MoMo obtains a higher average return than ModuMorph, consistently throughout the different environments. Similarly, R-MeMo performs better than MetaMorph, and in some environments and parameter changes even outperforms ModuMorph. These results demonstrate improved generalization performance obtained by the recurrent methods.

## 5.4 ZERO-SHOT GENERALIZATION TO UNSEEN ROBOT TOPOLOGIES

Gupta et al. (2022) constructed training and test robot topologies, $\mathcal{C}_{\text{train}}$ and $\mathcal{C}_{\text{test}}$, for which $\mathcal{C}_{\text{train}} \cap \mathcal{C}_{\text{test}} = \emptyset$. Even though these robots are sampled from the same distribution, they have a variable

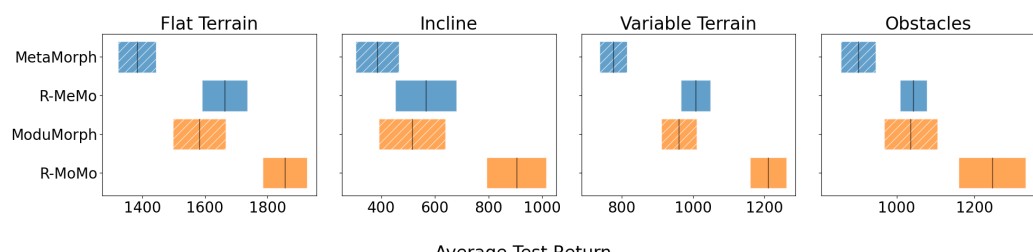

Figure 6: Test performance on robots with unseen topologies. The average return with a 95% confidence interval is shown.

amount of limbs (up to 12), with different context features, and a variable amount of joints per limb (up to 2), and are therefore extremely diverse. After training on the 100 training robots, the methods were evaluated on the 70 test robots with unseen topologies. The averaged returns of the different methods in the four environments are shown in Figure 6. In each of the environments, ModuMorph and/or R-MoMo dominate training, but R-MoMo significantly outperforms the other methods in zero-shot generalization to the test robots. Additionally, R-MeMo is competitive with ModuMorph on the unseen test robots, even though ModuMorph performs better during training. In Appendix C, experiments are reported with less robots in the training set, showing a consistent zero-shot generalization improvement of the recurrent architecture also when having access to fewer training robots. These results show that the recurrent architectures can learn policies that generalize much better to unseen test robots than their non-recurrent baselines. Note that the gap between training and test performance remains substantial by comparing the average returns of Figures 4 and 6, indicating that generalization remains challenging.

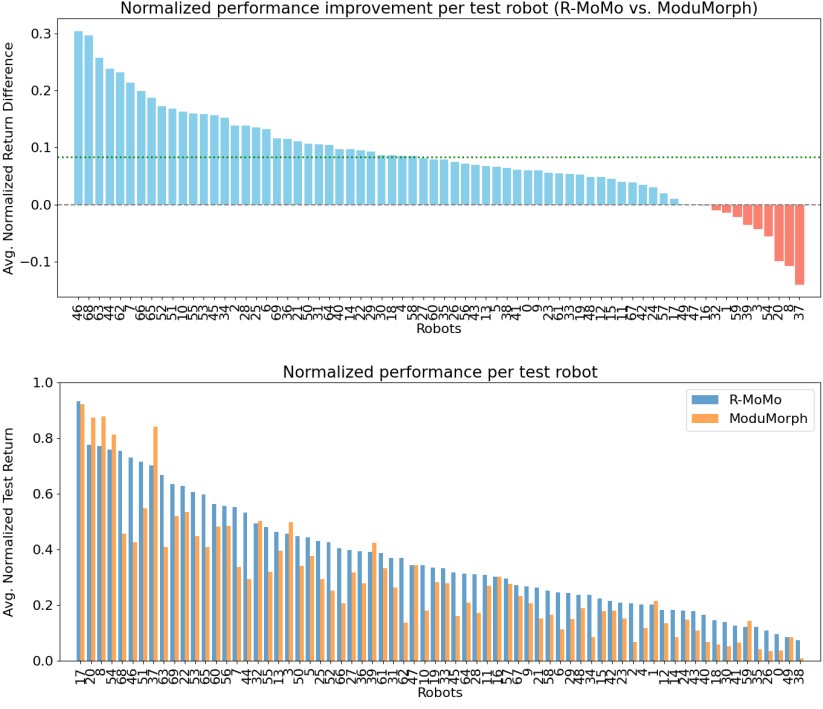

Figure 7: The difference in return between R-MoMo and ModuMorph (top) and the obtained returns (bottom) on each of the 70 unseen test robots. The returns are normalized for each environment and averaged over 10 seeds per environment. The green dotted line indicates the average performance improvement over all test robots.

Returns in each environment can range from very low ($< 0$) to very high ($> 4000$) values. Averaged returns over 70 test robots could therefore be misleading, as differences can be caused by only a small set of test robots. We therefore additionally report the difference in return between R-MoMo and Modumorph for every test robot individually, next to the obtained returns, in Figure 7. To compare the methods across all environments, the average returns are normalized using the minimal $(52, -74, 71, 126)$ and maximal $(5008, 3751, 3000, 2403)$ returns found in Flat Terrain, Incline, Variable Terrain, and Obstacles, respectively. Per-robot performance comparisons for every environment separately can be found in Appendix C. This comparison shows a consistent improvement for a majority of the test robots, illustrating the increased zero-shot generalization performance. Additionally, robots that ModuMorph performs better on are often also well controlled by R-MoMo. In contrast, ModuMorph struggles to control various robots across environments, where R-MoMo shows substantially less robots that are poorly controlled (i.e. with a very low return).

## 5.5 SINGLE CONTEXTUAL FEATURES

We experimented with a scenario in which a very limited number of contextual features would be available to the agent, to find potential differences in robustness against this lack of information. In these experiments, we provide ModuMorph and R-MoMo with only a single contextual feature and evaluate performance on training and unseen test morphologies. The contextual features are described in Table 1 in Appendix A. The average return after training, compared to the scenario where *all available* contextual features are provided, is shown in Figure 8 for the Flat Terrain and Incline environments. Note that large error bars (e.g. for ModuMorph with body_mass in Flat Terrain) correspond to highly instable performance across different random seeds. These results indicate that only single contextual features can already result in reasonably good training and testing performance. In contrast, specifically in Incline, which is more difficult than Flat Terrain, we can clearly see that some features do not provide enough information for good control. Interestingly, ModuMorph's test performance seems to be higher when only provided with body_ipos as compared to getting all available context features, which is not the case for R-MoMo for any of the individual features. These results suggest that R-MoMo can better infer context that is relevant for the task from a set of features in which only some are informative.

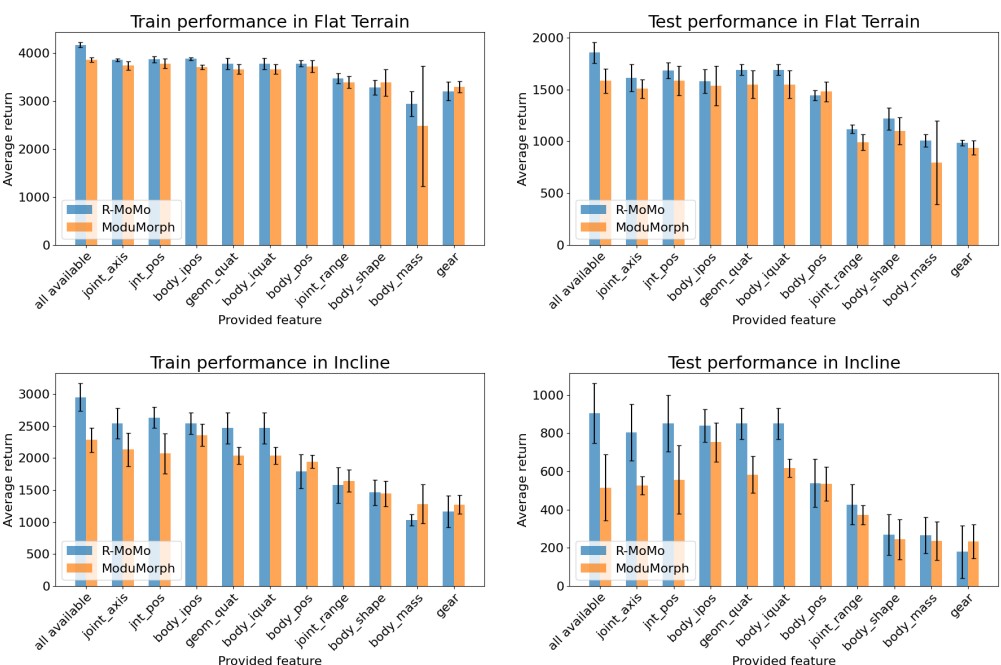

Figure 8: Final training (left) and test (right) performance of ModuMorph and R-MoMo on Flat Terrain (top) and Incline (bottom) when providing either *all available* contextual features or only a single contextual feature. Mean performance and standard deviation over 5 seeds is shown.

## 6 DISCUSSION

This work explores the introduction of modular recurrence in transformer-based architectures for improved multi-robot control. It was hypothesized that modular recurrence could allow the agent to infer relevant unobservable context to improve performance. The results have shown a consistent increase in zero-shot generalization performance when such memory-mechanism was introduced across different environments for robots with different dynamics, kinematics, and topologies. This clearly indicates that the RNN extracts *some* unobservable context information from the history.

Experiments in which only a single context feature is provided did not consistently show increased performance for all features. However, specifically in a more difficult environment, informative context features can enable the recurrent architecture to learn better policies, indicating that the implicit inference of more contextual information is dependent on the quality or informativeness of available context features. This is an interesting observation, as it implies that learning of unobservable contexts greatly benefits from observing some features, but not others. We leave it to future work to characterize what kind of features assists this inference best.

A limitation of the explored recurrent architecture, is that hidden states have to be stored for each limb. Scaling to robots with a large number of limbs requires, therefore, more memory. Besides, the sequential processing of RNNs results in longer training times (although this can be minimized by efficient batch processing through episode chunking). An interesting direction for future research would be to investigate a more efficient memory-mechanism in the architecture. Lastly, the gap between training and test performance is still large and allows for further investigation. Nonetheless, the combination of modular recurrence with transformers has shown to be promising for multi-robot control and could be effective in other problems with graph-like structures.

## 7 REPRODUCIBILITY STATEMENT

The repository that contains the code with the implementation of the methods, and with which all experiments can be reproduced, can be found here: `[anonymized for review]`.

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

## A  OBSERVATIONS AND CONTEXT

The observation provided to the agent in the current environments consists of various features for each limb. Table 1 lists all these features with a short description, the dimensionality of the feature and whether the feature is part of the context. We refer to the MuJoCo documentation (Todorov et al., 2012) for more details.

Table 1: Features of observations and context with description and dimensionality. $*$ indicates that each limb contains this feature twice, as every limb can contain two joints.

| Feature | Description | Dim. | Context |
|---------|-------------|------|---------|
| body_xpos | Current x,y,z position of each limb | 3 | No |
| body_xvelp | Linear velocity of each limb | 3 | No |
| body_xvelr | Angular velocity of each limb | 3 | No |
| body_xquat | Orientation of each limb | 4 | No |
| qpos | Generalized coordinates of each joint | 1* | No |
| qvel | Generalized velocity of each joint | 1* | No |
| body_pos | Initial x,y,z position of limb w.r.t. parent limb | 3 | Yes |
| body_ipos | Initial x,y,z position of center of mass w.r.t. parent | 3 | Yes |
| body_iquat | Inverse quaternion of limb orientation | 4 | Yes |
| geom_quat | Quaternion of geom relative to the body | 4 | Yes |
| body_mass | Limb mass | 1 | Yes |
| body_shape | Limb shape | 2 | Yes |
| jnt_pos | Initial (x,y,z) coordinate of each joint | 3* | Yes |
| joint_range | Range of motion (lower and upper bound) of each joint | 2* | Yes |
| joint_axis | Axis of rotation/translation of each joint (one-hot for x,y,z) | 3* | Yes |
| gear | Gear ratio for each joint | 1* | Yes |

## B  HYPERPARAMETERS AND COMPARISONS

### B.1  REGULARIZATION

In our experiments, we use the same hyperparameter values as in MetaMorph and ModuMorph. We only evaluate two different values for a regularization parameter on the validation set. Xiong et al. (2023) namely argued that this parameter can have a big impact on performance. This parameter defines the maximum approximate KL-divergence between the old and the new policy for every mini-batch before the update step. If this value is exceeded, the iteration of updates ends, and we return to sampling new trajectories. Figure 9 shows the average performance of the different methods with the two validated values (3 and 5) that were also evaluated in ModuMorph. In most cases, there is not a big difference in performance. For every method, the value that resulted in the highest average return on the validation set was used for the reported results in Section 5.

We additionally experimented with two versions of recurrent experience replay for R-MeMo and R-MoMo, in which we either reset hidden states at the start of each stored chunk in the roll-out buffer, or store the initial hidden state of each chunk. The latter showed a more stable training performance and was therefore used. This is to be expected, because a zero initialization at every chunk requires the agent to recover a meaningful hidden state during the burn-in period, in which it can only depend on transitions from the roll-out buffer. We therefore report results with stored hidden states.

### B.2  INCREASED PARAMETERS FOR BASELINES

Since the recurrent architectures come with an increased number of trainable parameters, we also experimented with versions of MetaMorph and ModuMorph with a similar number of additional trainable parameters. For MetaMorph, the first embedding block is simply increased by two fully connected layers (with a hidden size of 256 and ReLU activation). In ModuMorph these layers were added after the first embedding block, where the RNN is placed in R-MoMo. Figure 10 shows that the extra trainable parameters causes instability issues during training. Therefore, we stick to the architectures with the original number of parameters for the other experiments reported in this paper.

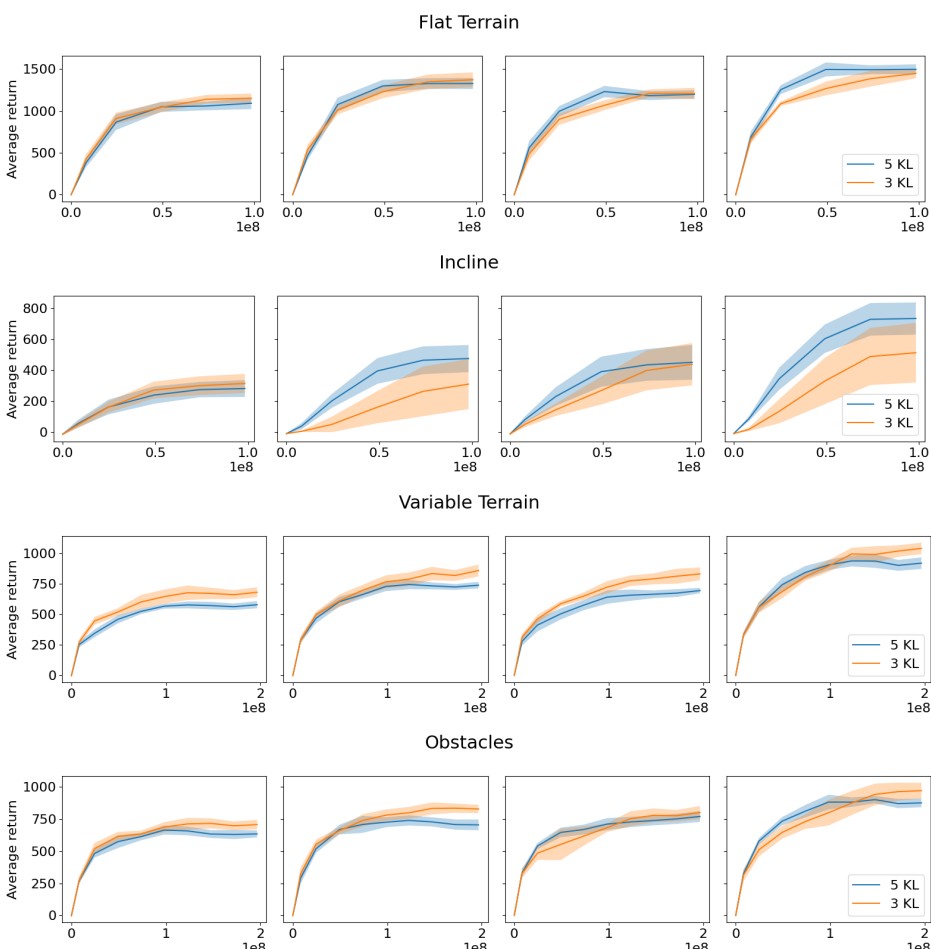

Figure 9: Validation performance on the 32 robots in the validation set for different values of the regularization parameter that defines the maximum approximate KL-divergence between the old and the new policy (e.g. 5 KL corresponds to a max. approximate KL-divergence of 5.0), averaged over 10 seeds with shown 95% confidence intervals.

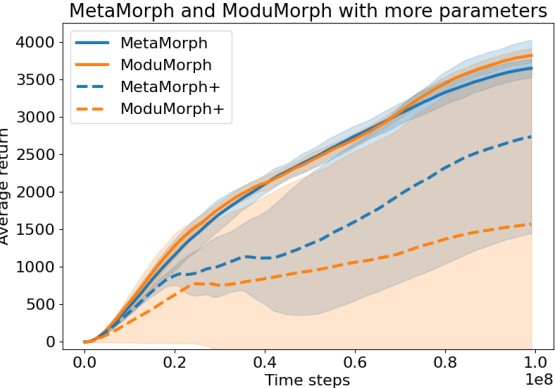

Figure 10: Training performance of MetaMorph and ModuMorph with additional trainable parameters (MetaMorph+ and ModuMorph+), comparable to the amount of added parameters in the recurrent architectures, in Flat Terrain. The average return over 5 seeds with 95% confidence intervals is shown.

### B.3 WALL CLOCK TIME

In Table 2 the wall clock time of the different methods for 100M environmental steps and training with the currently used hyperparameter values in Flat Terrain is reported (note that this includes logging to different destinations and storing model checkpoints). Experiments were performed on Nvidia A40 GPUs. Chunking the collected episodes before training, and applying the burn-in strategy introduced by Kapturowski et al. (2018) for DRQN, minimizes the extra overhead of the RNN's sequential processing, but still results in a significant increased training time. The trade-off between performance and the reduction of the chunk and/or burn-in size could be further exploited to reduce training time, which we leave for future work.

Table 2: The wall clock time of the different methods for 100M time steps of training on Flat Terrain, averaged over 10 seeds.

|  | Wall clock training time |
| --- | --- |
| MetaMorph | $16.3 \pm 0.5$ hours |
| R-MeMo | $30.0 \pm 0.7$ hours |
| Modumorph | $21.9 \pm 0.6$ hours |
| R-MoMo | $33.6 \pm 1.1$ hours |

## C ZERO-SHOT GENERALIZATION PERFORMANCE COMPARISON

### C.1 ARCHITECTURE ABLATION

Instead of inserting the RNN in the base controller, it can also be introduced in the hypernetwork to learn to infer unobservable contextual information. In this case, we add the AOH representation to the mapped observable context, to form the input for the blocks that generate the embedding layer, the query and key, and the decoder layer. The original RNN in the base controller (i.e. before the Transformer) is removed. We refer to this variation as HyperR-MoMo and illustrate it in Figure 11. Note that in R-MoMo the hypernetwork output can be computed only at the start of an episode, since the context does not change during an episode. HyperR-MoMo requires a forward pass at every time step. The training performance of this architecture variation in Flat Terrain is shown in Figure 12,

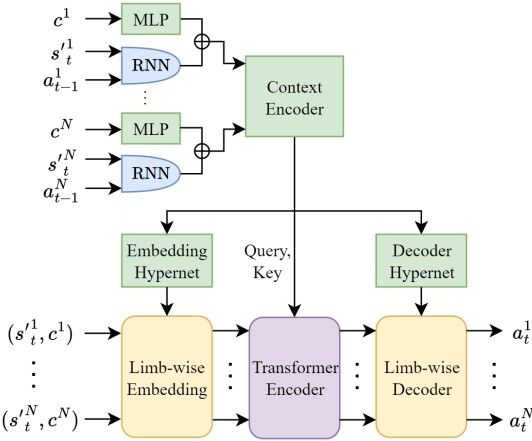

Figure 11: Illustration of the HyperR-MoMo architecture variation, in which the AOH representation is learned in the hypernetwork instead of in the base network.

and its zero-shot generalization performance in Figure 13, comparing it with the original ModuMorph and R-MoMo architectures. In Flat Terrain, notably, HyperR-MoMo obtains significantly lower training performance, even though its average test performance is between ModuMorph and R-MoMo. In Incline, HyperR-MoMo shows the lowest training performance, and a similar test performance as ModuMorph. The lower performance and large variance across runs is due to a few runs in which the agent did not learn at all. These results indicate that this architecture variation might require additional hyperparameter optimization, but could be still be promising for universal morphology control.

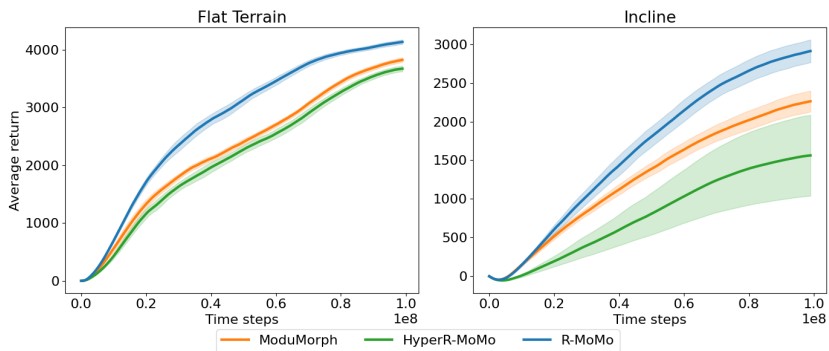

Figure 12: Training performance on the 100 training robots in Flat Terrain of ModuMorph, R-MoMo, and HyperR-MoMo. Average return with a 95% confidence interval over 10 seeds is shown.

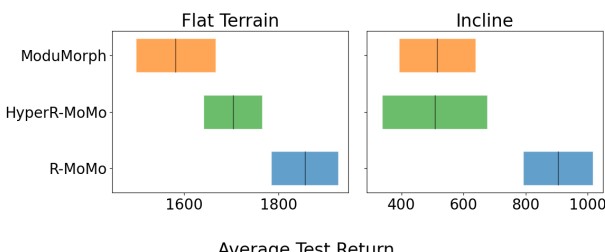

Figure 13: Return obtained by ModuMorph, R-MoMo, and HyperR-MoMo (modified R-MoMo with RNN in hypernetwork) on the 70 unseen test robots in Flat Terrain. Averages over 10 seeds with 95% confidence intervals are denoted.

## C.2 TRAINING SET SIZE

Figure 14 shows the return obtained by the different methods for different training set sizes of 25, 50, 75, and the complete set of 100 training robots in Flat Terrain and Incline. The improvement in zero-shot generalization performance of R-MoMo over ModuMorph is consistently visible for the different sizes of training sets; the difference seems to increase when more training robots are available.

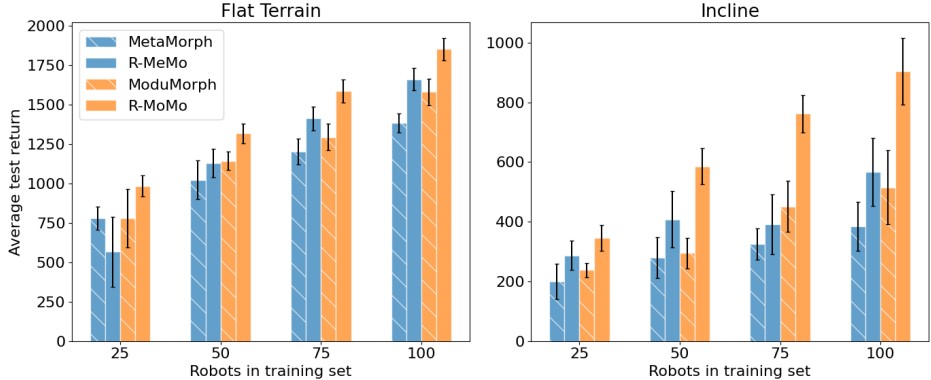

Figure 14: Return obtained on the 70 unseen test robots in Flat Terrain and Incline for smaller training set sizes (25, 50, and 75 are randomly sampled from the total of 100 training robots), averaged over 10 seeds with 95% confidence interval error bars.

## C.3 PERFORMANCE IMPROVEMENT PER ENVIRONMENT

The difference in performance on test robots between R-MoMo and ModuMorph is shown in Figures 15, 16, 17 and 18 for the Flat Terrain, Incline, Variable Terrain and Obstacles environments, respectively. These results show increased test performance for R-MoMo on a majority of the test robots across all environments. Moreover, R-MoMo generally struggles with fewer robots (e.g. return below 500) than ModuMorph.

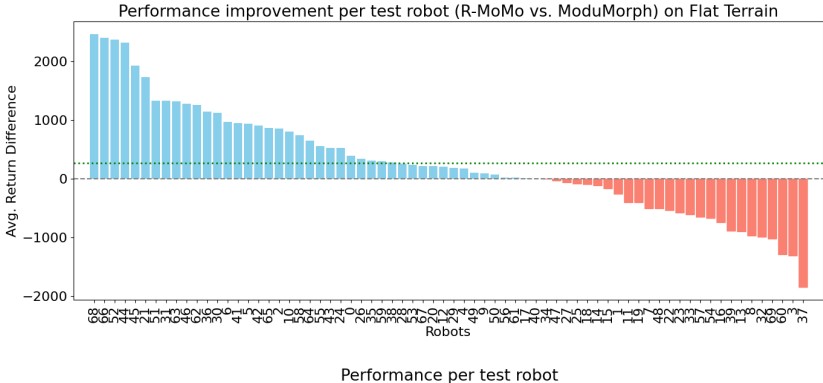

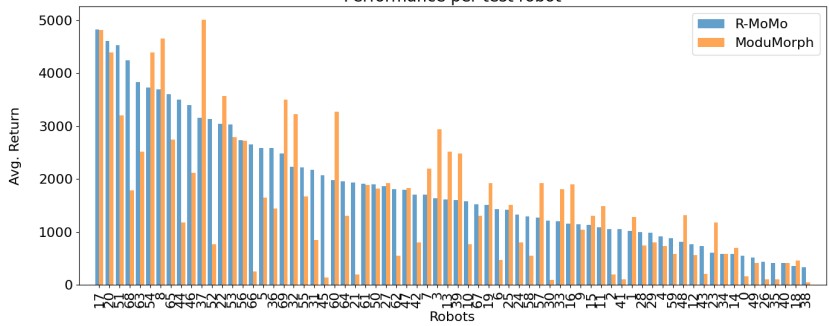

Figure 15: The difference in return between R-MoMo and ModuMorph (top) and the obtained returns (bottom) on each of the 70 unseen test robots in the **Flat Terrain** environment. Returns are averaged over 10 seeds.

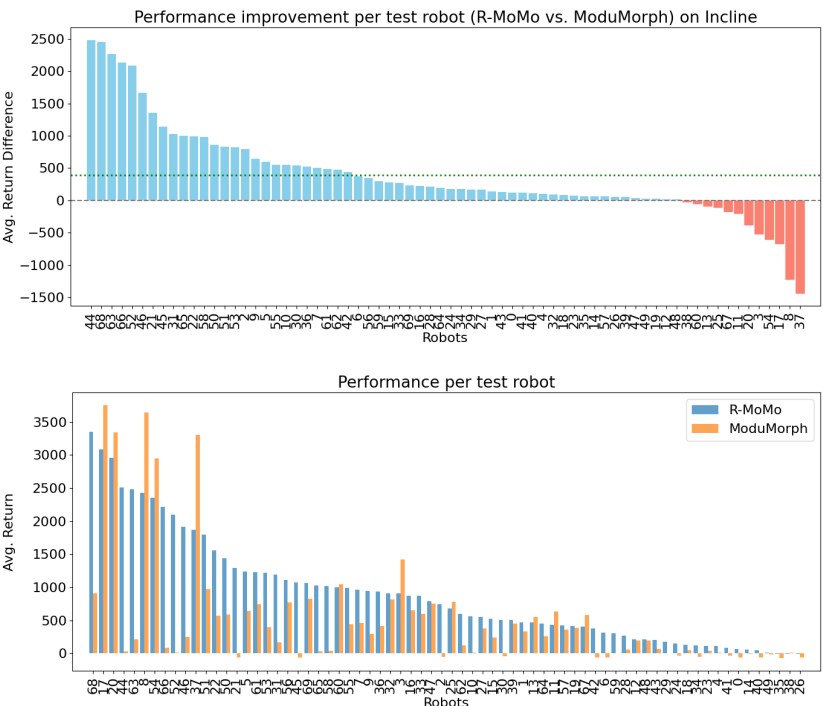

Figure 16: The difference in return between R-MoMo and ModuMorph (top) and the obtained returns (bottom) on each of the 70 unseen test robots in the **Incline** environment. Returns are averaged over 10 seeds.

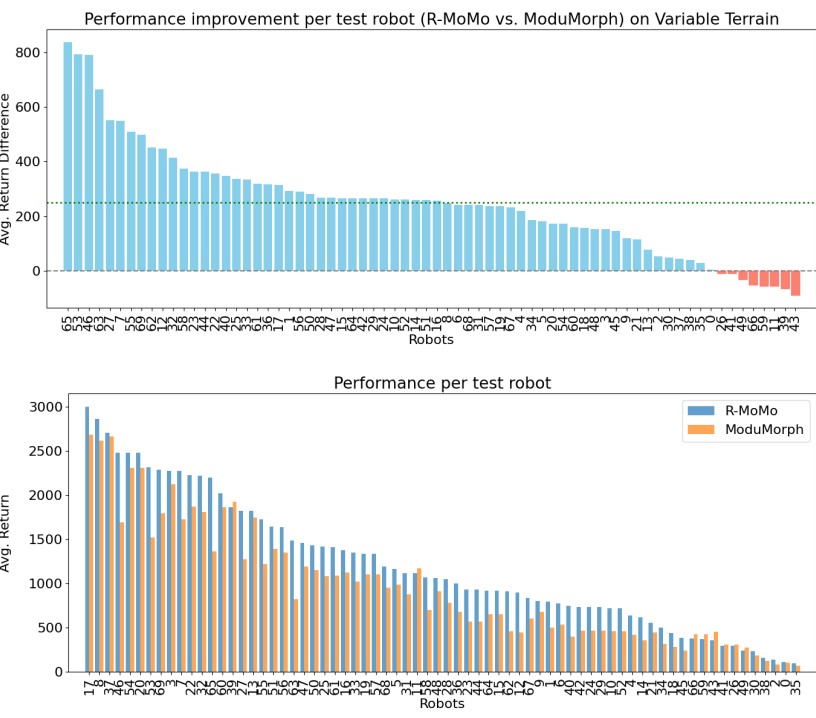

Figure 17: The difference in return between R-MoMo and ModuMorph (top) and the obtained returns (bottom) on each of the 70 unseen test robots in the **Variable Terrain** environment. Returns are averaged over 10 seeds.

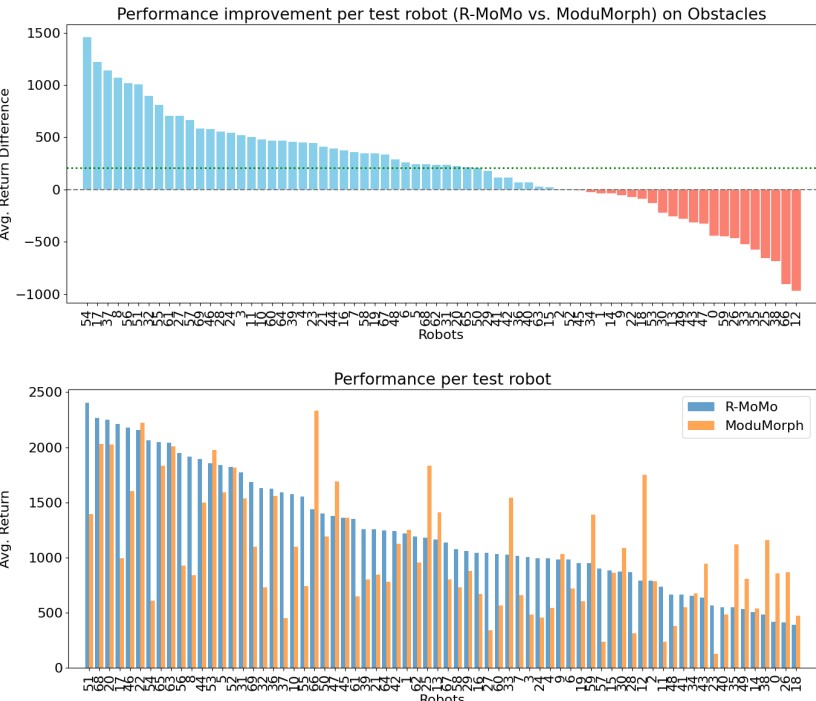

Figure 18: The difference in return between R-MoMo and ModuMorph (top) and the obtained returns (bottom) on each of the 70 unseen test robots in the **Obstacles** environment. Returns are averaged over 10 seeds.

