# OpenReview forum: "Shared Modular Recurrence for Universal Morphology Control"
_ICLR.cc/2026/Conference — Submitted to ICLR 2026_

### Official Review · Reviewer_jmJo · 2025-10-29

**Soundness:** 2
**Presentation:** 3
**Contribution:** 2
**Rating:** 4
**Confidence:** 3

**Summary:**

This paper builds upon previous modular robotic control approaches, and proposes to infer unobservable but relevant contextual information from history interactions using recurrent networks to enhance cross-embodiment generalization. The resulting modular recurrent transformer-based architectures, R-MeMo and R-MoMo, are validated across four commonly adopted MuJoCo environments, yielding notable performance gains compared with original networks without recurrence.

**Strengths:**

1.This paper addresses an important problem in robotic control, i.e., learning universal controllers generalizable to morphologically different agents. The motivation of inferring contextual information from environmental interactions is interesting (though I respectfully believe this motivation is not fully supported by the experiments; Please see Weakness 1).

2.The experimental results are promising, outperforming the latest baselines by a large margin. Four simulation environments with varying difficulty levels are examined, increasing the credibility.

3.The authors provide detailed research background and related works, clearly delineating the relationships between their work and the literature.

**Weaknesses:**

1.One of the key claims of this paper is that some unobservable contextual features could be inferred from environmental interactions. However, the notable performance drops seen when some critical features are removed, as reported in Figure 8, indicate that much of the contextual features could not be successfully recovered, which seems contradictory.

2.Since the proposed methods, R-MeMo and R-MoMo, largely build upon MetaMorph and ModuMorph, the authors are suggested to provide a more detailed introduction to their architectures, in order not to cause confusion in readers not familiar with these prior works.

**Questions:**

1.The use of RNN for dealing with POMDP seems a common practice. Following Weakness 1, how could one eliminate the possibility that the modular recurrence is merely learning to recover unobservable state transitions (as in a standard POMDP setting) rather than morphological contexts? I would be happy to raise my rating if the authors could by some means disentangle these two, for example, by showing correlation between RNN representations and morphological features.

2.Could the authors explain why, in the R-MoMo architecture, AOH is fed into the base controller (i.e., Transformer) rather than into the context encoder and used for generating network parameters alongside the observable contexts?

---

> ### Author Response · Authors · 2025-11-19
>
> We want to thank the reviewer for their time and appreciate their careful consideration of our work. We are glad that the reviewer recognizes the importance of learning universal controllers and that they consider our experimental results to be promising, outperforming state-of-the-art baselines. We address the reviewer's concerns below, and want to specify here that we have added new experiments in Appendix B.2, C.1, and C.2 on basis of the reviews. All adjustments and additions to the paper have been marked as red text in the revised PDF.
>
> > One of the key claims of this paper is that some unobservable contextual features could be inferred from environmental interactions. However, the notable performance drops seen when some critical features are removed, as reported in Figure 8, indicate that much of the contextual features could not be successfully recovered, which seems contradictory.
>
> Figure 8 shows, indeed, that the recurrent method still relies on the availability of relevant contextual features. This can be caused, however, by different aspects, e.g., the fact that the architecture is still heavily reliant on contextual features (the hypernetwork conditions only on context), or that some individual contextual features do simply not contain enough information about the robot morphologies to allow generalizable control. The aim of this experiment was to identify differences in an (extremely) limited context setting. We do observe that in the Incline environment the recurrent versions outperform their baseline for roughly half of the features (up to $\mathrm{body\\_iquat}$), both in training and testing, which would be expected if the agent can infer more context.
>
> > Since the proposed methods, R-MeMo and R-MoMo, largely build upon MetaMorph and ModuMorph, the authors are suggested to provide a more detailed introduction to their architectures, in order not to cause confusion in readers not familiar with these prior works.
>
> We thank the reviewer for pointing out this potential confusion for readers. We have adjusted Section 4.2 to introduce the most important aspects of the baseline architectures and discuss our extensions accordingly.
>
> > The use of RNN for dealing with POMDP seems a common practice. Following Weakness 1, how could one eliminate the possibility that the modular recurrence is merely learning to recover unobservable state transitions (as in a standard POMDP setting) rather than morphological contexts? I would be happy to raise my rating if the authors could by some means disentangle these two, for example, by showing correlation between RNN representations and morphological features.
>
> We thank the reviewer for this question, as it concerns a core aspect of our argument. In the CMDP setting with multiple robots introduced in section 2.2, the state can be decomposed into the underlying state $s_t'$ of the environment and the context $c$, which describes the robot to be controlled. In the single-robot setting, with a single constant context $c$, the state in MuJoCo environments is usually considered to be fully observable, i.e. $o_t = s_{t}'$. In the multi-robot setting, on the other hand, the agent still observes the full environmental state $s'_t$, but potentially only a part of the full context $o_t = (s'_t, c^+), c^+ \subset c$. The only unobservable state features are therefore in the context. We agree with the reviewer that the modular recurrence can recover unobservable state transitions, but if MuJoCo is indeed fully observable, the unobservable features can only correspond to morphological context features. Any recovering of unobservable state transitions in our experiments must therefore recover morphological contexts. We added a clarification of this to Section 2.2.
>
> > Could the authors explain why, in the R-MoMo architecture, AOH is fed into the base controller (i.e., Transformer) rather than into the context encoder and used for generating network parameters alongside the observable contexts?
>
> Thank you very much for pointing this out. The suggested way of exploiting the AOH is a valid alternative. We hypothesized that having AOH information more explicitly in the latent embeddings would better enable the Transformer to learn relationships between limbs. Besides, the suggested variation would require a forward pass through the hypernetwork at every time step of the episode. In R-MoMo, instead, the hypernetwork output can be computed only at the start of each episode, since the context does not change during an episode. We have, however, added an ablation experiment on basis of the reviewer's question, in which the RNN is part of the hypernetwork instead of the base controller. Specifically, the RNN in this case encodes the AOH and forms, together with the mapped available context features, the input for the hypernetwork blocks that generate the embedding layer, query and key of the transformer, and decoder layer.
>
> The added ablation can be found in Appendix C.1.

---

### Official Review · Reviewer_18Nq · 2025-10-31

**Soundness:** 2
**Presentation:** 3
**Contribution:** 2
**Rating:** 6
**Confidence:** 3

**Summary:**

The paper proposes addressing the challenging problem of training a single Reinforcement Learning policy to control any robot morphology. The authors hypothesise that key robot characteristics (like friction or damping) are partially observable introduce a per-limb Recurrent Neural Network (RNN) into modular transformer-based architectures, demonstrating a substantial and consistent improvement in generalizing to robots.

**Strengths:**

* The use of RNN for the partially observable CMDP is well-motivated in Section 2.2. It would be even nicer to provide a few quantitative evidence to show the partial observability.
* Empirically, the work successfully demonstrates a significant and consistent increase in generalisation performance to unseen robot morphologies, dynamics, and kinematics over strong baselines (MetaMorph and ModuMorph).

**Weaknesses:**

* While the paper hypothesises that recurrence allows the agent to infer specific unobservable context (like friction or damping), the paper does not include an explicit analysis or visualisation that confirms what the RNN is encoding or how well it correlates with the true (unobservable) physical properties.
* The authors also comment on the slow training speed of the RNN. It would be helpful to provide the training speed of the experiments. And also some ablation studies on the key hyperparameters, e.g. RNN's latent state, shared network.

**Questions:**

1. For R-MoMo and R-MeMo, why are the positions of the RNN different? Can they inserted both before the embedding or before the transformer?
2. Can you briefly comment on the training and testing robots in Section 5.4? Are they sampled from the same distribution?
3. We know that RNN is an architecture to solve a Meta RL problem. Are there other advanced Meta RL methods helpful to this case?

---

> ### Author Response · Authors · 2025-11-19
>
> We thank the reviewer for their time and constructive engagement with our paper. We are glad that the reviewer found our proposed method well-motivated, and that they recognize the significant and consistent increase in generalization performance upon state-of-the-art baselines. Their questions and concerns are discussed below. We want to point out that we have added new experiments in Appendix B.3, C.1, and C.2. on basis of all received reviews. All changes and additions to the paper are marked as red text in the revised PDF.
>
> > While the paper hypothesises that recurrence allows the agent to infer specific unobservable context (like friction or damping), the paper does not include an explicit analysis or visualisation that confirms what the RNN is encoding or how well it correlates with the true (unobservable) physical properties.
>
> We initially designed the experiments in Section 5.5 for this purpose, to show that the RNN can infer missing context features from observations. In particular in the Incline environment of Figure 8 one can see that the RNN architectures perform clearly better when only one of the 10 features is available. However, for some features (starting with body pos) the RNN was not able to infer some crucial context and both training and test performance reduce to their baselines. We do not know what makes these particular features especially hard to infer, but see the improvement by RNN in half of the tested cases as quantitative evidence for our hypothesis.
>
> We agree with the reviewer that more explicit analyses or visualizations would strengthen this line of thought. However, it is complicated to design such analyses since it is not known which (unobservable) context features are relevant for the task. Particularly, for different limbs the agent might need to infer different context features for optimal control.
>
> > The authors also comment on the slow training speed of the RNN. It would be helpful to provide the training speed of the experiments. And also some ablation studies on the key hyperparameters, e.g. RNN’s latent state, shared network.
>
> We added training speed comparisons for the methods we experimented with in Appendix B.3. Ablation studies and hyperparameter optimization for the RNN were not performed, due to computational limitations. We stuck to the same dimensionality for the hidden states as the size of the embeddings in the original MetaMorph [1] and ModuMorph [2] architectures. We consider it an advantage of the proposed modifications that they did not require additional, extensive hyperparameter tuning.
>
> [1] A. Gupta, et al. “MetaMorph: Learning universal controllers with Transformers.” International Conference on Learning Representations, 2022.
>
> [2] Z. Xiong, et al. “Universal morphology control via contextual modulation.” International Conference on Machine Learning, 2023.
>
> > For R-MoMo and R-MeMo, why are the positions of the RNN different? Can they inserted both before the embedding or before the transformer?
>
> For R-MoMo: inserting the RNN before the embedding layer results in larger inputs to the embedding layer (if we keep the size of the hidden states the same): 128 instead of 54 per limb. For each limb, the hypernetwork would then generate ($128 \times 128 =$) 16384 parameters, instead of ($54 \times 128 =$) 6912. In turn, the embedding hypernetwork itself would now have ($128 \times 16384 =$) around 2M trainable parameters, instead of ($128 \times 6912 =$) almost 900k. Thus, for a fair comparison with ModuMorph, we inserted the RNN after the embedding layer. We added this clarification to the manuscript add the end of Section 4.2.
>
> For R-MeMo one could experiment with the RNN after the embedding layer, but we chose the current implementation to be able to separate actions and observations from the available context, which is static during an episode. We did not consider the roughly 10k increase in trainable parameters for the shared embedding MLP significant, and, besides, experimented with larger embedding MLPs in the baseline MetaMorph in Appendix B.2.
>
> > Can you briefly comment on the training and testing robots in Section 5.4? Are they sampled from the same distribution?
>
> Gupta et al. [1] constructed a set of training robot topologies, $\mathcal{C}\_{\mathrm{train}}$, and test robot topologies, $\mathcal{C}\_{\mathrm{test}}$ for which $\mathcal{C}\_{\mathrm{train}} \cap \mathcal{C}\_{\mathrm{test}} = \emptyset$. Even though these robots are sampled from the same distribution, they have a variable amount of limbs (up to 12), with different context features, and a variable amount of joints per limb (up to 2), and are therefore extremely diverse. We added this comment to the manuscript at the beginning of Section 5.4.

---

> > ### Author Response · Authors · 2025-11-19
> >
> > > We know that RNN is an architecture to solve a Meta RL problem. Are there other advanced Meta RL methods helpful to this case?
> >
> > We thank the reviewer for this suggestion. In this paper we look into zero-shot generalization of the learned policies, but one could interpret access to the preceding trajectory also as few-shot adaptation to the test environment. In this light, gradient-based meta-learning methods like MAMAL [1] could be employed to adapt the network after every step in the environment. However, this would slow down execution considerably in comparison to RNN inference. Another option would be in-context learning [2, 3], in particular with transformers [4]. This is very similar to using RNN, as the transformer would also have access to the agent's history. However, here too, RNNs are faster during execution as they are linear in the length of the history, not quadratic as transformers. We added this discussion to the new related works section 3.3.
> >
> > [1] C. Finn et al. "Model-Agnostic Meta-Learning for Fast Adaptation of Deep Networks." International Conference on Machine Learning, 2017.
> >
> > [2] A. Moeini et al. "A Survey of In-Context Reinforcement Learning." https://arxiv.org/abs/2502.07978, 2025.
> >
> > [3] Y. Duan et al. "RLˆ2: Fast reinforcement learning via slow reinforcement learning." https://arxiv.org/abs/1611.02779, 2016.
> >
> > [4] M. Laskin et al. "In-context Reinforcement Learning with Algorithm Distillation" International Conference on Learning Representations, 2023.

---

> > > ### Comment · Reviewer_18Nq · 2025-11-26
> > >
> > > Thanks for answering my questions.

---

### Official Review · Reviewer_yQpS · 2025-11-01

**Soundness:** 3
**Presentation:** 2
**Contribution:** 2
**Rating:** 2
**Confidence:** 4

**Summary:**

This paper builds upon previous transformer-based universal control methods (MetaMorph and ModuMorph) by introducing a recurrent model to handle partially observable robot contexts. These robot contexts include properties like robot limb mass, shape, gear ratio for each joint, etc, as listed in the appendix. The authors hypothesize that the relevant contextual information can be partially observable, and in such cases, the added recurrent module allows the agent to infer hidden contextual information, achieving better multi-robot control and better generalization to unseen robot contexts. The experiments on MuJoCo show consistent improvements in zero-shot generalization across unseen robot morphologies, dynamics, and kinematics, demonstrating that integrating recurrence helps the controller adapt more effectively to diverse and unfamiliar robots.

**Strengths:**

- The performance gain is very consistent, showing promising benefits of the proposed shared modular recurrence.
- Though I find simple and incremental, this work presents a very clear motivation and well-defined problem setting.

**Weaknesses:**

- The scope of the experiments seems very limited to me considering the large diversity of possible robot morphologies. The paper does not analyze how the recurrent mechanism scales with larger or more diverse sets of robots, nor does it investigate the relationship between the amount of training data and the observed generalization gains. Without experiments on different dataset sizes or more complex morphological distributions, I'm not convinced that the proposed recurrent module truly improves generalization in a scalable and robust manner.
- The proposed approach mainly extends existing transformer-based frameworks MetaMorph and ModuMorph by introducing a recurrent layer to handle partial observability, which is also a hypothesis brought by this work. Although this modification leads to performance improvements, it represents a relatively minor architectural change without introducing new learning principles or theoretical insights. The paper positions the work as addressing partial observability, but the underlying idea of adding recurrence to capture temporal dependencies is conceptually straightforward and has been widely explored in prior reinforcement learning and robotics studies. Therefore, the novelty and contribution may not be sufficient.
- Several figures use inconsistent evaluation scales, which makes it difficult to visually compare performance across different settings, especially for training/testing comparison. For example, in Figure 4 the returns in Incline reach nearly 3000 during training, whereas in Figure 6 the corresponding test returns drop below 1000.

**Questions:**

- Are there any other methods beyond MetaMorph and ModuMorph that can be integrated with recurrence?
- Why in Fig. 8, train and test performance in Flat Terrain, the variance of ModuMorph when provided with body_mass is exceptionally large? Besides, the x-axis labels in Fig. 8 are slightly misaligned.
- Why do you think making the context of a robot partially observable is important? To my understanding, the context of a control task is usually available.

---

> ### Author Response · Authors · 2025-11-19
>
> We thank the reviewer for their time and thoughtful suggestions. We are glad that the reviewer finds our motivation clear, problem setting well-defined, and our proposed method to show promising benefits. Below we discuss the reviewer’s concerns and questions, and refer to changes and additions to the paper based on the reviewer’s comments. Please let us know if we missed any or if new ones arise. Additional experiments on basis of the received reviews can be found in Appendix B.3, C.1, and C.2. All changes and additions to the paper are marked as red text in the revised PDF.
>
> > The paper does not analyze how the recurrent mechanism scales with larger or more diverse sets of robots
>
> In line with the literature [1,2], we emphasize that the UNIMAL design space contains a huge variety of robot morphologies ([1] estimate that the UNIMAL design space contains more than $10^{18}$ robot morphologies): every robot can differ in action space, observational input, morphology, and dynamics, since the number of limbs, how those are connected, and their contextual properties can be different for each robot. MetaMorph [2] introduced a training (100) and testing (100) set of robots that can contain up to 12 limbs, with a maximum of two joints per limb, and demonstrated that it is (still) very challenging to learn a universal controller that can generalize to robots it has not seen during training.
>
> [1] A. Gupta, et al. “Embodied intelligence via learning and evolution.” Nature Communications, 2021.
>
> [2] A. Gupta, et al. “MetaMorph: Learning universal controllers with Transformers.” International Conference on Learning Representations, 2022.
>
> > nor does it investigate the relationship between the amount of training data and the observed generalization gains.
>
> Thank you for this suggestion. An ablation experiment was added to the paper in Appendix C.3, in which we show zero-shot generalization performance to unseen robot topologies for different numbers of training robots (25, 50, 75, and the original total training set size of 100). The results show a consistent improvement of the proposed modular recurrent architecture across all training set sizes.
>
> > The proposed approach mainly extends existing transformer-based frameworks [. . . ] Although this modification leads to performance improvements, it represents a relatively minor architectural change without introducing new learning principles or theoretical insights
>
> The scope of this paper is to (1) identify the gap in literature for universal morphology control, where it was always (in our view) unjustifiably assumed that all relevant information about robot morphologies is directly accessible, and (2) to combine existing methodologies, such as e.g. modular policies, recurrence, shared parameters, recurrent experience replay, and hypernetworks, into a novel method to improve performance of a universal controller that can be trained effectively on only a limited set of robots.
>
> > Several figures use inconsistent evaluation scales, which makes it difficult to visually compare performance across different settings
>
> We agree that the challenge of morphology-agnostic control is far from being solved. Due to the complexity of the task of learning a universal controller, the generalization gap remains substantial in state-of-the-art methods (as discussed in the conclusion). Especially in more complex environments, such as Incline, Variable Terrain and Obstacles. Therefore, we, and previous works such as MetaMorph and ModuMorph, report training and testing performance in separate figures, to be able to reason about performance differences in these different settings individually. We added a clarifying sentence in the manuscript (Section 5.4, L401-403) to avoid confusion about differences between performance in different settings.
>
> > Are there any other methods beyond MetaMorph and ModuMorph that can be integrated with recurrence?
>
> Since recurrence can ideally learn to identify the node-contexts, it can be integrated into any modular method to learn morphology-agnostic control, for example Amorpheus [3] or SMP [4].
>
> [3] V. Kurin, et al. “My body is a cage: the role of morphology in graph-based incompatible control.” International Conference on Learning Representations, 2021.
>
> [4] W. Huang, et al. “One policy to control them all: shared modular policies for agent-agnostic control.” International Conference on Machine Learning, 2020.

---

> > ### Author Response · Authors · 2025-11-19
> >
> > > Why in Fig. 8, train and test performance in Flat Terrain, the variance of ModuMorph when provided with body mass is exceptionally large? Besides, the x-axis labels in Fig. 8 are slightly misaligned.
> >
> > Thank you for pointing this out. In this experiment, we leave out 9 out of 10 context features to identify differences in performance between ModuMorph and R-MoMo. When only body mass is available, ModuMorph seems to learn a very poor policy in at least one of the random seeds, which causes a large variance in the observed returns. We clarified this in the manuscript (Section 5.5, L452-453). We also adjusted Figure 8 so that the x-axis labels are aligned as expected and the legend does not overlap with the error bars, and changed the label for the scenario where all available features are provided to the agent from all to all available to avoid readers to think that we can provide all contextual features (which we cannot because we do not have access to all features).
> >
> > > Why do you think making the context of a robot partially observable is important? To my understanding, the context of a control task is usually available.
> >
> > We do not artificially make the context partially observable, we argue that every set of contexts to differentiate the limbs and joints in a robot is inherently heuristic and can always leave out some crucial context feature. We demonstrate this by showing that a modular recurrent architecture, which can identify at least some of the context from interactions alone, improves both training and test performance in state-of-the-art architectures. We consider this strong evidence, although not conclusive proof, that some important context features must be unobservable in these standard benchmarks.

---

### Author Response · Authors · 2025-11-26

Hereby, we would like to let the reviewers know that we have added more experimental results to the paper. To be able to make more robust statements about the added experiments in Appendix C.1 (architecture ablation) and C.2 (training set size ablation), we have now included results on the Incline environment. Additionally, for the experiment in C.2, we added results for the other methods, MetaMorph and R-MeMo, that were missing in the previous version.

---

### Meta-Review · Area_Chair_J775 · 2026-01-08

**Summary:**

The paper proposes Shared Modular Recurrence, an extension to modular transformer-based universal morphology control architectures that introduces per-limb recurrent neural network (RNN) to infer relevant unobservable contextual information  (e.g., friction, damping, limb interactions) from observation histories. Building on MetaMorph and ModuMorph, the proposed method introduces recurrent variants of both architectures (R-MeMo and R-MoMo) using shared limb-level RNNs, and evaluates them on a large and diverse set of MuJoCo robots drawn from the UNIMAL design space across four locomotion environments. Experimental results demonstrate consistent improvements in both multi-robot training performance and, more notably, zero-shot generalization to robots with unseen dynamics, kinematics, and topologies.  ￼

The paper was reviewed by three referees, who broadly agree that the proposed solution is well-motivated and that the experimental results reveal consistent performance gains over the baselines, demonstrating the advantages of modular recurrence. At the same time, the reviewers raised questions/concerns about the limited scope of the experimental evaluation, notably the limited diversity in the robot morphologies that are considered, the lack of any analyses of the dependence on the amount of training data, and the need for ablation studies. Reviewers comment that these issues make it difficult to determine whether the method would improve generalization for a broad class of robot morphologies. Additionally, at least one reviewer finds that the paper would benefit from an explicit analysis of what contextual information the RNN iis inferring and how it correlates with unobserved physical properties. Other concerns include questions about training efficiency and  inconsistent scales used in plots, which makes it difficult to compare results.

**Reviewer Concerns:**

In their rebuttal, the authors made a concerted effort to address these concerns. The authors added new experiments that evaluate performance as a function of training set size, that provide additional architectural ablations, and training-speed comparisons, arguing that the recurrent models consistently outperform their non-recurrent counterparts. Additionally, the authors clarified design choices regarding RNN placement in R-MeMo and R-MoMo, and expanded the discussion that situates recurrence relative to meta-learning and in-context learning alternatives. With respect to partial observability, the authors emphasized that robot context is inherently heuristic and incomplete in practice, and pointed to improved performance under reduced context as empirical, though indirect, evidence that recurrence helps to infer missing information. While acknowledging that more direct analyses of learned representations would strengthen the claims, they argue that such analyses are nontrivial given the ambiguity of which latent context variables matter for control.

**Reviewer Scores:**

The authors' responses helps to strengthen the paper and should be incorporated in any future version of the paper. However, it is not clear that the clarifications and additional results/discussions would sufficiently sway the reviewers.

---

### Decision · Program_Chairs · 2026-01-26

Reject